# Humility and Competence: Which Attribute Affects Social Relationships at Work?

**DOI:** 10.3390/ijerph19105969

**Published:** 2022-05-14

**Authors:** Ai Ni Teoh, Livia Kriwangko

**Affiliations:** School of Social and Health Sciences, James Cook University, Singapore 387380, Singapore; liv.kriwangko@gmail.com

**Keywords:** humility, competence, likability, work environment, seniority

## Abstract

Between likability and competence, people value likable colleagues (regardless of their competence level) more than competent colleagues. If humility replaces competence, the preference might be different since humility is not always associated with positive outcomes. Humility and competence form four archetypes: humble star, humble fool, competent jerk, and incompetent jerk. This study examined the personal and professional preferences for these archetypes in the workplace and how the preference is moderated by colleagues’ seniority. There were 475 working adults aged between 21 and 77 (*M* = 40.34, *SD* = 11.32) recruited to complete an online survey. While humble fools were more likable than competent jerks in personal interactions, competent jerks received more cooperation than humble fools in professional interactions. Seniority did not affect these findings. Our findings shed light on whether, and when, humility should be highly valued in organizational settings. Promoting humility in the workplace setting might require more caution.

## 1. Introduction

Collaboration and teamwork are essential in organizational settings. Hence, researchers have been investigating factors affecting interpersonal interactions in the workplace [1]. For instance, Casciaro and Lobo [2] suggested that likability and competence affect interpersonal interactions. Similar to likability, humility is an attractive trait that is frequently studied and highly valued in organizational settings [3]. It is intuitive to expect that humility and competence may affect interpersonal interactions in the same way as likability and competence. However, the evidence does not always link humility to positive outcomes (e.g., [4]). Therefore, in the present study, we replaced likability with humility in Casciaro and Lobo’s model, to examine how humility and competence affect interactions. The findings will have implications for the contexts when humility or competence should be valued in organizational settings.

### 1.1. The Likability–Competence Model

Researchers have proposed several models to explain how the traits of colleagues affect interpersonal interactions. Among these models, Fiske et al. [5] and Fiske [6] focused on warmth and competence dimensions, whereas Casciaro and Lobo [2] focused on likability and competence dimensions. While Fiske et al.’s model is generally applicable to all interpersonal interactions, Casciaro and Lobo’s model is applicable to workplace interactions.

Casciaro and Lobo’s [2] model suggests that working with likable teammates makes interactions more pleasant. Moreover, since teamwork usually involves task completion and performance, working with competent teammates is necessary to complete tasks efficiently. The dimensions of likability and competence form four archetypes [2]. The first is the lovable star, who is highly likable and highly competent. The second is the lovable fool, who is highly likable but incompetent. The third is the competent jerk, who is unlikable but highly competent. The last archetype is the incompetent jerk, who is unlikable and incompetent. See Figure 1 for the likability–competence model. While the first two archetypes are clearly welcomed due to their likable trait, and incompetent fools are not due to their unlikable trait, whether competent jerks or lovable fools are more welcome is not immediately clear. 

In the study by Casciaro and Lobo [2], employees from four diverse organizations rated their colleagues in terms of how much they liked the colleague and how good the colleague was at the job. Results from the study showed that employees preferred working with lovable fools compared to competent jerks. According to the authors, being likable plays a bigger role than being competent in work relationships [2]. Choosing competent jerks over lovable fools may be necessary for a professional decision, but working with competent jerks is not easy. They have unpleasant attitudes and may keep information to themselves or seek selfish gains. Lovable fools, on the other hand, are more pleasant to work with. They are more willing to share the information and skills they have, contributing without self-centered motivations [2]. As such, employees gravitated towards working with lovable fools.

### 1.2. How about the Humility–Competence Model?

Likability broadly refers to positive self-presentation, physical attractiveness, compliments, and association [7]. Humility, on the other hand, is a specific concept, considered as a positive trait that is highly valued in society [8]. It has gradually received research attention in relation to its applicability to organizational settings, management, and leadership (see [3] for a review). However, given that humility has not always been associated with positive outcomes (e.g., [4,9,10]), promoting humility in the workplace setting might require more caution. In the present study, we explored how the humility of colleagues affects the outcomes of interactions. As such, we replaced “likability” in Casciaro and Lobo’s [2] model with humility and examined the *humility*–competence model.

Humility involves concepts such as accurately assessing one’s strengths and weaknesses, acknowledging one’s mistakes and limitations, willingness to learn, appreciating others, and having a low self-focus [11]. It is the midpoint between arrogance and low self-esteem. Possessing humility allows individuals to distinguish the fine line between healthy self-confidence and over-confidence [12]. Humble individuals are willing to improve, by being open to suggestions and criticism and, hence, are generally likable [8].

When we introduce humility into Casciaro and Lobo’s [2] model to replace likability, the four archetypes become humble stars, humble fools, competent jerks, and incompetent jerks. The preference for these four archetypes may be different. Although we expect humble stars to remain the most popular archetype and incompetent jerks the least, two possible compensatory effects might occur when it comes to the preference between humble fools and competent jerks. 

First, when considering humility and competence, the compensatory effect of humility might occur, where humility compensates for a lack of competence. That is, humble fools are preferred over competent jerks. In a study by Owens and colleagues ([13], Study 2), they found a significant interaction effect between undergraduate students’ general mental ability (a variable analogous to competence) and expressed humility on their individual performance on a subject. As expected, competent jerks and humble stars did not differ in performance. However, competent jerks did not perform significantly better than humble fools. It appears that the humility trait of humble fools compensates for their lack of competence because they are more receptive towards feedback, knowing their own weaknesses and others’ strengths [13]. While the compensatory effect of humility occurs at the personal level (i.e., individual performance), the effect may also occur at an organizational level. Organizations that manifest humility in their practices are more likely to have outstanding performance as humble behaviors facilitate innovation through an open attitude of experimentation and discussion [12]. In other words, interpersonal relationships based on humility can also be an asset that positively affect organizational performance. 

Apart from performance, indirect evidence has shown that having a humble character may compensate for a lack of competence in interpersonal interactions [10]. Humble individuals make team members feel secure to voice ideas and suggestions, without worrying about receiving disrespectful remarks [1,14]. Having a humble leader generates more team information sharing and facilitates a psychologically safe environment, both interpersonally [15] and within the team [1,14]. This positive influence is especially optimized when there is consistency in the leader’s humility [16]. In sum, a humble trait may compensate for competence, where humble fools are preferred over competent jerks.

The second possible outcome is the compensatory effect of competence, where competence compensates for the lack of humility, and competent jerks are preferred over humble fools. Working with people who are not humble may be less enjoyable [2]. However, when an arrogant colleague has competence to offer, s/he will improve team performance. Therefore, competence may compensate for a lack of humility, especially in a workplace setting where performance and goal attainment are highly regarded. 

Competence not only compensates for a lack of humility, but it may also compensate for possessing the trait of humility. Humility is often associated with self-humiliation, harsh self-criticism [10], low self-esteem, and self-deprecation [11]. When a leader, who people naturally expect to be competent and dominant, demonstrates a humble trait, people will see s/he lacks authority [10]. In this case, competence might compensate for having the trait of humility, because it is advantageous for goal attainment. The compensatory effect of competence might lead to a preference for competent jerks, particularly in situations where competence is highly sought after.

### 1.3. The Context-Dependent Compensatory Effects

The compensatory effects of humility and competence depend on the context of interaction. In workplace settings, colleagues may interact with each other at a personal level (e.g., lunch) and at a professional level (e.g., discussing projects). Person-level interactions do not require competence, and hence competence holds little importance in such contexts. The compensatory effects of humility may occur in person-level interactions where people prefer interacting with humble fools. On the other hand, professional interactions require competence to facilitate the completion of tasks. Therefore, the compensatory effects of competence may occur in professional interactions where people prefer interacting with competent jerks. 

In peer-to-peer interactions in the workplace, we might interact with seniors—who have more working experience, knowledge, and expertise than us—and juniors [17]. Interacting with seniors or juniors may affect our preference for the four archetypes. People expect juniors to be respectful and obedient [18], implying a compensatory effect of humility. When a junior colleague demonstrates humility and open-mindedness, people are more inclined to collaborate and share information [19]. Conversely, seniors may view competent juniors as competition and thus offer less sharing of information and collaboration [20]. On the other hand, people generally tolerate humble or arrogant responses from their seniors because the hierarchy discourages speaking up [18]); thus, opting to engage in negative gossip instead [20]. Therefore, it seems that colleagues place more emphasis on seniors’ competence than on their humility, implying a compensatory effect of competence.

### 1.4. Aims and Hypotheses

The present study aimed to examine the preference for the four archetypes in work relationships. Since “preference” is a broad term, our study focused on personal likability, professional likability, and how much cooperation participants gave to colleagues. Of these variables, personal likability applies to personal relationship. Professional likability and cooperation are task-related variables, with professional likability referring to the attitude toward colleagues, and cooperation focusing on actions related to collaboration. 

We predicted that humble stars would receive the highest ratings in all outcome variables, and incompetent jerks the lowest. Between humble fools and competent jerks, we predicted that the compensatory effect of humility would take place when it comes to relationship-related rating, where people would find humble fools more personally likable. However, a compensatory effect of competence may take place when it comes to task-related ratings, where people would find competent jerks more professionally likable and provide more cooperation with them. 

In addition, we examined the moderating effect of seniority. We predicted that the compensatory effect of humility would occur when participants interact with junior colleagues, and the compensatory effect of competence would occur when participants interact with senior colleagues.

## 2. Materials and Methods

### 2.1. Participants and Design

The present study is a correlational study. We recruited 450 working adults from Prolific, a crowdsourcing platform, and 25 through social media in Singapore, making the total sample size 475. We used two sampling approaches because we needed Eastern and Western samples to examine the moderating effect of collectivism, one of our initial study plans. Respondents on Prolific were generally based in the United Kingdom and United States, and, therefore, we recruited Western participants via this platform. Convenience sampling in Singapore may reach Eastern participants. However, since we did not have a sufficient sample size for Eastern participants, we did not examine the moderating effect of collectivism. Independent-sample *t*-tests showed that the participants recruited via the two approaches were comparable (see Section 3.2). Therefore, we combined the data for the main hypothesis testing. See Table 1 for the characteristics of the sample. 

Our participants were aged between 21 and 77 (*M* = 40.34, *SD* = 11.32). Most of them were women and were from the United Kingdom and the United States of America. Participants’ work experience ranged from less than a year to 58 years. On average, they had had about 20 years of working experience (*M* = 19.96, *SD* = 11.54). Out of the 475 participants, 233 were seniors, 206 were juniors, 21 had the same years of experience as their collaborators, and 15 did not indicate an answer.

### 2.2. Materials

We administered the following scales in a randomized order, except for the recall task, which was administered first, and the demographic scale, which came last. We also randomized the order of the items within each scale. 

#### 2.2.1. Recall Task

Participants were instructed to recall a peer with whom they had recently collaborated on a project at work. Based on this recalled colleague, participants wrote a brief description of the person and the project in less than 80 words.

#### 2.2.2. Relational Humility Scale

The 16-item relational humility scale [21] measures an observer’s judgment of another person’s humility level. For the purpose of the study, we replaced “he/she” with “my collaborator” in each item. The sample item includes “*My collaborator* knows his/her strengths.” Participants rated each item on a five-point Likert scale, from 1 (*Strongly disagree*) to 5 (*Strongly agree*), based on the collaborator recalled. A past study [21] showed that the scale has good construct validity. The Cronbach’s Alpha of this scale was 0.93 in the present study.

#### 2.2.3. Perceived Competence Scale

The perceived competence scale was developed by Williams and Deci [22]. We adapted all four items to fit the purpose of the study, for instance “I feel confident in my ability to learn the relevant materials” was changed to “*My collaborator* felt confident in *his/her* ability to learn the relevant materials.” Participants rated each item on how competent they perceived their collaborators to be on a seven-point Likert scale from 1 (*entirely disagree*) to 7 (*entirely agree*). In a previous study [23], the scale yielded acceptable internal consistency and factorial validity. The scale had a high internal consistency (α = 0.90) in the present study.

#### 2.2.4. Reysen Likability Scale

This is an 11-item scale developed by Reysen [7], measuring how likable participants perceived their collaborators to be (e.g., “This person was warm”). Participants rated each item on a seven-point Likert scale 1 (*very strongly disagree*) to 7 (*very strongly agree*). The reliability and validity of this scale were high [24]. The scale also had a high internal consistency of 0.94 in the present study. 

#### 2.2.5. Willingness to Cooperate Scale

The 5-item willingness to cooperate scale [25] measures willingness to cooperate with someone using a five-point scale, from 1 (*strongly disagree*) to 5 (*strongly agree*). We replaced the words “other employees” from the original scale with “my collaborator” to fit our study purpose (e.g., “I was willing to enhance communication with *my collaborator* on the project”). Participants rated how strongly they agreed with each statement based on the collaborator recalled. The internal consistency and validity of this scale were high [26]. The Cronbach’s alpha for this scale was 0.88 in the present study.

#### 2.2.6. Demographic Scale

This scale asked participants for information, including age, gender, and the country they are based in, as well as their work experience and their collaborator’s work experience in years. 

After reverse scoring items, where necessary, we averaged all the ratings of each scale to obtain average scores, with higher scores indicating a higher level of the construct measured. To obtain the seniority score, we deducted collaborators’ years of work experience from participants’ years of work experience. Positive scores on seniority indicate participants being senior to their collaborators, and negative scores indicate participants being junior.

### 2.3. Procedure

After receiving ethics clearance from the University’s Human Research Ethics Committee (approval number H7983), we published the study on Prolific and social media to recruit participants. Interested and qualified participants clicked the link that directed them to our online survey on Qualtrics. The survey first showed the information sheet of the study and an informed consent form, followed by “agree” and “disagree” buttons. Those who hit the “disagree” button indicating dissent were directed to exit the survey. Those who selected the “agree” button indicating consent were directed to the recall task. 

## 3. Results

### 3.1. Assumption Check

We found several univariate outliers in the sample and capped the outliers at 3.29 standard deviations from the mean. The normality assumption was violated in all variables. Nevertheless, we did not perform data transformation because our sample size was large and, hence, robust against violation of the assumption when performing MANOVAs and multiple regression [27]. A multivariate outlier was removed from the dataset (final *N* = 474). The homoscedasticity assumption was satisfied. We tried hypothesis testing with two sets of data, with and without outliers. Keeping or capping the outliers did not affect the results of the hypothesis testing, and, therefore, the results reported below are based on the dataset with outliers capped.

### 3.2. Preliminary Analyses

Independent-sample *t*-tests showed that participants recruited from Prolific and social media did not differ in terms of the study variables. We also performed correlation analyses, to examine the associations among the study variables and demographic variables. See Table 2 for the correlation coefficients.

### 3.3. Likability Construct

Although not tested in the original scale validation paper [7], the 11-item Reysen likability scale seems to comprise eight items related to personal likability (liking a person for their personal qualities) and three items related to professional likability (liking a person for their professional qualities). Therefore, we conducted a confirmatory factor analysis (CFA) using IBM SPSS AMOS 27 to test the two-factor structure of likability. The initial model indicated a poor fit, χ^2^(43) = 356.07, *p* < 0.001, χ^2^/df = 8.28, TLI = 0.91, CFI = 0.93, RMSEA = 0.12. We removed two items from personal likability (“I would like this person as a roommate” and “This personal was physically attractive”) due to low regression coefficients (βs < 0.60) and correlated four pair of residuals. The fit statistics of this new model finally showed a good fit, χ^2^(22) = 50.94, *p* < 0.001, χ^2^/df = 2.32, TLI = 0.99, CFI = 0.99, RMSEA = 0.05. Figure 2 shows the final CFA model.

Based on the final CFA model, we computed the mean scores of personal likability and professional likability. Each of the variables had four univariate outliers, which were later capped at 3.29 standard deviations from the mean. The distributions of these variables were not normal. 

### 3.4. Hypothesis Test

#### 3.4.1. Main Effects

We performed a *k* means cluster analysis using IBM SPSS Statistics for Windows (v.27), IBM Corp, Armonk, NY, USA to form clusters of participants based on the humility and competence levels of their collaborators. Both the standardized humility, *F*(3, 470) = 375.23, *p* < 0.001, and competence variables, *F*(3, 470) = 473.26, *p* < 0.001, contributed significantly to the cluster formation. The first cluster was the biggest, with 217 participants who worked with humble stars; the second largest cluster comprised 123 participants who collaborated with humble fools; the third cluster had 97 participants who worked with competent jerks; the fourth and smallest cluster had 37 participants who worked with incompetent jerks. 

We performed a MANOVA using IBM SPSS Statistics (v.27) by entering the clusters as the independent variable (IV) and personal likability, professional likability, and cooperation as the dependent variables (DVs). The results showed a significant main effect for clusters on overall work relationships, *F*(9, 1139.14) = 50.40, *p* < 0.001, partial η^2^ = 0.238, Wilks’ Lambda = 0.442. Univariate analyses showed that the main effect was significant on personal likability, *F*(3, 470) = 121.43, *p* < 0.001, partial η^2^ = 0.437, professional likability, *F*(3, 470) = 103.21, *p* < 0.001, partial η^2^ = 0.397, and cooperation, *F*(3, 470) = 103.03, *p* < 0.001, partial η^2^ = 0.397. 

Multiple comparisons using Bonferroni correction showed that humble stars were the most likable personally, followed by humble fools, competent jerks, and incompetent jerks. Professionally, people liked humble stars most and incompetent jerks least, but they did not find humble fools and competent jerks to be different. When it comes to cooperation, participants gave the greatest cooperation to humble stars, followed by competent jerks, humble fools, and incompetent jerks. Table 3 tabulates the means and standard deviations of each DV for the four clusters.

#### 3.4.2. The Moderating Effects of Seniority

We conducted a series of moderation analyses, with the clusters as the categorical IV (competent jerk was compared with each of the remaining clusters) and seniority as the moderator. Moderation analyses provided the interaction effects between each pair of categories (e.g., competent jerk vs. humble star) and seniority. Therefore, each analysis produced three interaction effects (e.g., competent-jerk-vs-humble-star by seniority interaction effect). See Table 4 for the regression coefficients of each interaction effect.

Contrary to the hypothesis, the seniority of collaborators did not moderate the associations between the archetypes of collaborators and work relationships, except for the competent-jerks-versus-incompetent-jerks comparison. The interaction effect between this comparison and seniority was significant on cooperation. The simple slope analysis by seniority showed that when participants were more senior to their collaborators (*b* = −1.53, *t* = −9.14, *p* < 0.001), they were more cooperative with their junior collaborators who were competent jerks than their junior collaborators who were incompetent jerks. Such a pattern was less evident (but still significant) when participants were junior to their collaborators (*b* = −1.05, *t* = −6.47, *p* < 0.001). See Figure 3 for the moderating effect of seniority on the effects of the four archetypes on cooperation.

## 4. Discussion

### 4.1. Key Findings

Our participants ranked humble stars the highest for personal likability, followed by humble fools, competent jerks, and incompetent jerks. In terms of professional likability, humble stars were preferred over humble fools, and competent jerks were preferred over incompetent jerks. However, humble fools and competent jerks were given comparable rankings. Lastly, participants gave the greatest cooperation to humble stars and the lowest to incompetent jerks, and gave more cooperation to competent jerks than humble fools. 

In summary, the compensatory effects of humility and competence took place in different interaction settings. In personal interactions, humility compensates for the lack of competence, as humble fools were reported to be more likable than competent jerks. In professional interactions, competence compensates for the lack of humility where competent jerks received more cooperation than humble fools. 

Seniority did not affect the compensatory effects of humility and competence described above. However, people were more likely to give more cooperation to unlikable colleagues if the colleagues were competent and give less cooperation if the unlikable colleagues were incompetent. Such a pattern was more evident when the colleagues were participants’ juniors. 

### 4.2. Theoretical Implications

Several models have used a two-dimensional approach to conceptualize personal attributes. In one model, communion and agency dimensions are used to explain social judgments [28]. In another model, warmth and competence dimensions describe personal attributes [5,6]. Similarly, Casciaro and Lobo [2] used likability and competence to predict relationships with colleagues at work. Despite the differences in the terms used and the aspects of social judgement proposed, these models seem to align with each other. 

When we judge others based on warmth/likability and competence dimensions, we admire those who are high in warmth and competence (or lovable stars) and show contempt to those low in warmth and competence (incompetent jerks; [29]). For those high in warmth but low in competence (lovable fools), we tend to show pity and offer help, showing that warmth compensates for a lack of competence [29]. For those low in warmth but high in competence (competent jerks), we tend to envy them and plan to initiate active harm, such as hostility and harassment [29]. Oftentimes, people view competence as irrelevant when positive interpersonal affect (warmth and likability) is missing [2]. In other words, when there is an outright dislike in a relationship, people simply do not approach a colleague for help or provide cooperation, even when their colleague possesses competence. 

In the present study, we examined the communion dimension using the humility construct and the agency dimension using competence. Differently from Casciaro and Lobo’s [2] findings, where likability played a bigger role than competence in general, our findings showed that humility plays a bigger role only in certain situations. Specifically, people prefer humble fools over competent jerks when personal interaction is concerned. However, when it comes to task-related variables, people prefer working with competent jerks. Most of our participants were from Western countries, which have a higher task-orientation than person-orientation [30,31,32]. As such, competence might play a more important role in task-related variables. Such findings might be different in a sample with participants with a higher level of collectivism, and where humility is valued and observed [33]. 

We also examined the compensatory effects of humility and competence in the context of colleague seniority. The effects observed above remained and were not affected by seniority. However, seniors seem to be less tolerant with juniors who are incompetent jerks than with juniors who are competent jerks. Our hypothesis related to seniority was not supported, which might be due to cultural background. Most of the participants in the present study were from Western countries. Western cultures have a lower power distance [34], where people see colleagues as having equal power. In those cultures, people work collaboratively and interdependently, regardless of seniority [35]. The findings might not be applicable to cultures with a greater power distance. In such cultures, seniority affects interpersonal interactions, for instance, juniors do not feel comfortable voicing their opinions to seniors [18,35]. 

The context-dependent characteristics of the humility–competence model are also found in the warmth–competence model. People tend to give more weight to warmth than competence when they evaluate distant others, and give more weight to competence when evaluating themselves and interrelated others, such as close friends [28]. In our study, participants evaluated their collaborators, people with whom they worked closely together and shared the consequences of their collaboration. In line with previous studies, we observed the compensatory effect of competence, where people give more weight to competence. To add to such findings, we found that such an effect was evident when it came to task-related variables.

### 4.3. Practical Implications

Should we emphasize the virtue of humility or competence in organizational settings? The present study findings suggested both. We should emphasize humility to create a workplace environment that facilitates communication. Participants in our study gave high ratings to humble stars and humble fools in terms of personal likability. The humble acknowledgement of own limitations [11] makes humble stars and humble fools less threatening to colleagues. Furthermore, humble stars and humble fools appreciate others [11] and are willing to share information [2], providing a comfortable ambience in the workplace for collaborators [15], as well as for team members to share their ideas and suggestions [1,14,16]. Therefore, humble stars and humble fools can create workplace harmony and facilitate communication [2]. 

Having a nonthreatening, comfortable ambience for communication, alone, is not sufficient. Without competitiveness, a team could suffer in performance. Humble stars and competent jerks may increase the competitiveness of the team. Our findings showed that people do not discredit the competence of competent jerks and will provide cooperation to them in a professional sense to obtain knowledge and skills for success. Competent people may contribute to team performance by increasing work efficiency and outcomes [36,37]. They may also motivate team members to be more competitive [38]. 

Therefore, a team should be diverse in terms of the archetypes of the members. We can make use of the compensatory effect of humility to create a comfortable ambience for communication. In addition, we can make use of the compensatory effect of competence to contribute to team performance and motivate other team members, such as humble fools and incompetent jerks, to improve their competitiveness. Although competent jerks may selfishly withhold information or produce conflict, humble stars and humble fools may buffer the conflict by creating a non-hostile environment. As such, leaders should assign tasks by considering the various contributions people of these archetypes may bring [2].

### 4.4. Limitations and Suggestions for Future Studies

Our study examined the social judgment of the four archetypes from the *humility*–competence model. However, the limitations of the study call for a careful interpretation of the findings. First, our sample might not be representative of the population, since we recruited via a crowdsourcing platform. Despite this, samples recruited from crowdsourcing platforms are more representative than those from convenience sampling [39,40]. Second, we studied peer relationships at work. The findings might not be applicable to other forms of work relationships, such as supervisor–supervisee relationships. For instance, since managers prefer competent subordinates to likable subordinates [2], the compensatory effect of humility found in the present study might be absent when managers evaluate their subordinates. In addition, since the humble trait of leaders may discredit leaders’ authority [10], the compensatory effect of humility might not be observed when subordinates evaluate their managers. Third, our Westerner-majority sample might limit the generalizability of the findings. The main findings and findings pertaining to the moderation effect of seniority might have been different if the sample was Easterner-majority. 

Lastly, the unbalanced cell sizes of the four archetypes might affect the power of the analyses. In our study, most participants reported working with humble stars and humble fools (both comprised 72% of the sample). Only a small proportion (28%) of participants worked with competent jerks and incompetent jerks. Such percentages could be due to positive bias of retrospective recall of affect. In a study by Ben-Zeev and colleagues [41], participants who did not suffer from any mental illness tended to report an exaggeratedly higher level of positive affect when asked to retrospectively rate their emotional intensity during the previous week. Our participants had to recall their collaboration experience retrospectively and might have given an exaggerated rating of how positive they felt about the collaboration. Therefore, they might have given higher-than-actual ratings of the humility and competence levels of their collaborators. Future research may investigate participants’ evaluation of their collaborators of ongoing projects, instead of previous projects. 

Although our findings shed light on how people evaluate colleagues of the four archetypes, our study did not measure the archetype of participants. It is interesting to examine how people of an archetype view those of other archetypes. As discussed above, humble fools might create a nonthreatening work environment for competent jerks to behave in a less hostile way. Such a speculation requires more research effort to investigate. 

## 5. Conclusions

The present study aimed to investigate the humility–competence model and shed light on whether the virtue of humility or competence should be emphasized in organizational settings. Our findings pointed to both humility and competence as important characteristics to be emphasized at work, depending on the context. We prefer humble fools in day-to-day personal interactions. In turn, we prefer competent jerks in professional interactions. Having a diverse team of members of the four archetypes could provide a balanced dynamic within a team.

## Figures and Tables

**Figure 1 ijerph-19-05969-f001:**
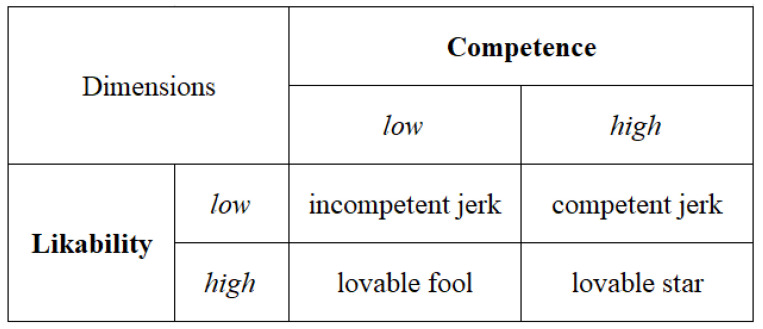
Likability–Competence Model [2].

**Figure 2 ijerph-19-05969-f002:**
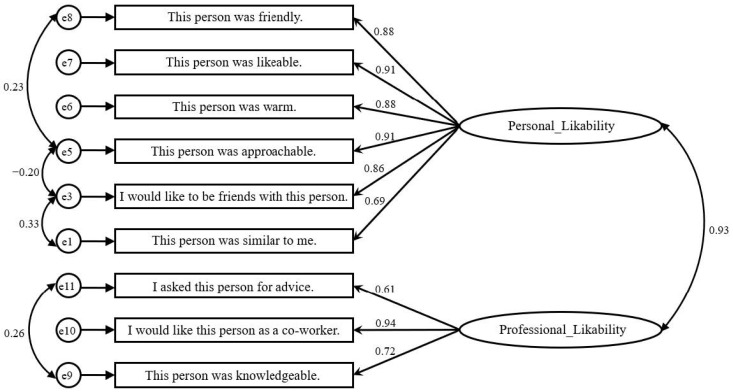
The CFA Model of the Construct of Likability.

**Figure 3 ijerph-19-05969-f003:**
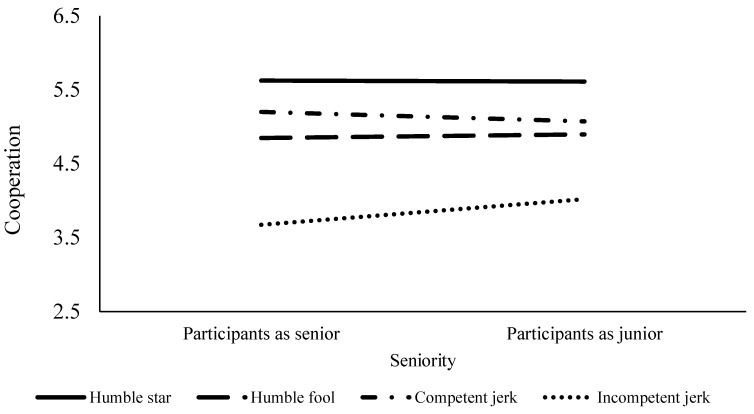
The Moderating Effect of Seniority on the Effects of the Four Archetypes on Cooperation.

**Table 1 ijerph-19-05969-t001:** The Demographic Characteristics of Participants.

Groups	*n*	Percentage
**Gender**		
Women	273	57.60%
Men	201	42.40%
**Country Based**		
Aust *	2	0.40%
Canada	1	0.20%
Hungary	1	0.20%
Indonesia	1	0.20%
Ireland	1	0.20%
Nort *	1	0.20%
Singapore	22	4.60%
Spain	2	0.40%
United Kingdom	280	59.10%
United States of America	163	34.40%

Note. *N* = 474 after assumption check. * Responses entered by participants.

**Table 2 ijerph-19-05969-t002:** Correlation Coefficient Among Variables.

	Age	Humility	Competence	Personal Likability	Professional Likability	Cooperation	Seniority
humility	0.000						
competence	0.070	0.484 **					
personal likability	0.009	0.750 **	0.573 **				
professional likability	−0.035	0.654 **	0.642 **	0.801 **			
cooperation	0.067	0.501 **	0.668 **	0.614 **	0.615 **		
seniority	0.493 **	−0.029	−0.010	0.022	−0.085	−0.008	
*M*	40.34	3.64	5.94	5.36	5.46	5.18	1.60
*SD*	11.32	0.78	1.00	1.21	1.14	0.79	12.11

Note. *N* = 474. ** *p* < 0.01.

**Table 3 ijerph-19-05969-t003:** Means and Standard Deviations of the Study Variables in Each Cluster.

Clusters	*n*	Personal Likability		Professional Likability		Cooperation	
		*M*	*SD*	*M*	*SD*	*M*	*SD*
humble star	217	6.11 ^a^	0.73	6.11 ^a^	0.71	5.61 ^a^	0.46
humble fool	123	5.15 ^b^	0.79	5.20 ^b^	0.82	4.87 ^c^	0.67
competent jerk	97	4.66 ^c^	1.14	5.05 ^b^	1.13	5.14 ^b^	0.74
incompetent jerk	37	3.49 ^d^	1.46	3.61 ^c^	1.25	3.86 ^d^	0.79

Note. The means in the same column with different superscripts differed significantly at *p* < 0.05.

**Table 4 ijerph-19-05969-t004:** The Regression Coefficients of the Interaction Effects.

Predictor	Moderator	Outcome Variables		
		Personal Likability	Professional Likability	Cooperation
competent jerks vs. humble stars	seniority	0.008	0.013	−0.005
competent jerks vs. humble fools	seniority	−0.003	0.003	0.007
competent jerks vs. incompetent jerks	seniority	0.020	0.016	−0.020 *

Note. * *p* < 0.05.

## Data Availability

Data are available from the corresponding author upon request.

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
