# Peer review of "Humility and Competence: Which Attribute Affects Social Relationships at Work?"

_ijerph, 2022, doi:10.3390/ijerph19105969_

Round 1

Reviewer 1 Report

I would like to thank the authors for this research that aims to examine the preference for the four archetypes in work relationships. Authors replaced likability with humility in Casciaro and Lobo’s model to examine how humility and competence affect interactions.

The research showed that the compensatory effects of humility and competence took place in different     interaction settings. In personal interactions, humility compensates for the lack of competence. In professional interactions, competence compensates for the lack of humility where competent jerks received more cooperation than humble fools.

The research subject is timely, innovative, and interesting. It also fits the aim and scope of the journal.

The research is well designed and follows a sound scientific research method.  Results and recommendations are clear and could have an impact among the community of researchers.

However, several adjustments are needed in order to improve the quality of the paper.

You need to adjust the abstract by adding two more key words like working environment as example.

Lines 42-47: It would be better and easier to present the traits under the format of a table.

Lines 68- 69 : you said: “In the present study, we focused on how humility of colleagues affects the outcomes of interaction”. You need to adjust the verb focused by another one more representative for the research like: we explored, we investigated.

There are serious concerns in regard to the section materials and methods:

How can you guarantee that your sample is representative and generates trustworthy results once you have only 475 respondents that come from around the world?

You did not explain why your respondents come from 2 different sources that are not equal (450 vs 25).

You did not explain why the majority (94%) of your respondents come from UK and USA and several other countries (6%)? What is the idea behind? Workers in UK and USA do not share same traits nor culture nor working values, so how could you justify your results especially once your research focuses on workers traits? Could this be highlighted as limitation?

Lines 218-219: In order to have a better idea about the impact of this variable (seniority), we need to know how many respondents were classified as senior and junior.

Lines 361-362: The important question that you need to answer, is why the cultural background was relevant once we talk about seniority, however it was not highlighted once discussion the other hypotheses?

Other minor comments are directly attached to the manuscript.

Reviewer 2 Report

This paper need improve the literature review that need more citation paper and new one. 

Round 2

Reviewer 1 Report

I would like to thank the authors for this research that aims to examine the preference for the four archetypes in work relationships. Authors replaced likability with humility in Casciaro and Lobo’s model to examine how humility and competence affect interactions.

The research showed that the compensatory effects of humility and competence took place in different     interaction settings. In personal interactions, humility compensates for the lack of competence. In professional interactions, competence compensates for the lack of humility where competent jerks received more cooperation than humble fools.

The research subject is timely, innovative, and interesting. It also fits the aim and scope of the journal.

The research is well designed and follows a sound scientific research method.  Results and recommendations are clear and could have an impact among the community of researchers.

The researchers made the necessary adjustments as suggested.